# Physiological Ovarian Aging Is Associated with Altered Expression of Post-Translational Modifications in Mice

**DOI:** 10.3390/ijms23010002

**Published:** 2021-12-21

**Authors:** Minli Wei, Jia Li, Huili Yan, Tao Luo, Jiang Huang, Yangyang Yuan, Liaoliao Hu, Liping Zheng

**Affiliations:** 1Nanchang University Jiangxi Medical College, Nanchang University, Nanchang 330031, China; 15579186101@163.com (M.W.); lijia4199@ncu.edu.cn (J.L.); yanyhl23@163.com (H.Y.); jian2004fei@126.com (J.H.); yyy2011030303@163.com (Y.Y.); 15970445860@163.com (L.H.); 2Key Laboratory of Reproductive Physiology and Pathology of Jiangxi Province, Nanchang University, Nanchang 330031, China; luotao@ncu.edu.cn; 3Institute of Life Science and School of Life Science, Nanchang University, Nanchang 330031, China

**Keywords:** physiological ovarian aging, histone modifications, female reproductive functions, post-translational modifications, mice

## Abstract

Post-translational modifications (PTMs) have been confirmed to be involved in multiple female reproductive events, but their role in physiological ovarian aging is far from elucidated. In this study, mice aged 3, 12 or 17 months (3M, 12M, 17M) were selected as physiological ovarian aging models. The expression of female reproductive function-related genes, the global profiles of PTMs, and the level of histone modifications and related regulatory enzymes were examined during physiological ovarian aging in the mice by quantitative real-time PCR and western blot, respectively. The results showed that the global protein expression of Kbhb (lysineβ-hydroxybutyryllysine), Khib (lysine 2-hydroxyisobutyryllysine), Kglu (lysineglutaryllysine), Kmal (lysinemalonyllysine), Ksucc (lysinesuccinyllysine), Kcr (lysinecrotonyllysine), Kbu (lysinebutyryllysine), Kpr (lysinepropionyllysine), SUMO1 (SUMO1 modification), ub (ubiquitination), P-Typ (phosphorylation), and 3-nitro-Tyr (nitro-tyrosine) increased significantly as mice aged. Moreover, the modification level of Kme2 (lysinedi-methyllysine) and Kac (lysineacetyllysine) was the highest in the 3M mice and the lowest in 12M mice. In addition, only trimethylation of histone lysine was up-regulated progressively and significantly with increasing age (*p* < 0.001), H4 ubiquitination was obviously higher in the 12M and 17M mice than 3M (*p* < 0.001), whereas the modification of Kpr (lysinepropionylation) and O-GlcNA in 17M was significantly decreased compared with the level in 3M mice (*p* < 0.05, *p* < 0.01). Furthermore, the expression levels of the TIP60, P300, PRDM9, KMT5B, and KMT5C genes encoding PTM regulators were up-regulated in 17M compared to 3M female mice (*p* < 0.05). These findings indicate that altered related regulatory enzymes and PTMs are associated with physiological ovarian aging in mice, which is expected to provide useful insights for the delay of ovarian aging and the diagnosis and treatment of female infertility.

## 1. Introduction

Ovarian aging is a complex biological process influenced by the interaction and gradual accumulation of internal and external factors [1,2], which can lead to a decline in estrogen levels, a shortened menstrual cycle, and a decrease in female reproductive capacity until the final menopause. It can also lead to a series of menopausal symptoms caused by endocrine disorders, such as cardiovascular diseases, osteoporosis, and obesity [3,4,5,6,7,8], which seriously affect women’s physical and mental health and quality of life. However, the commonly used clinical diagnostic indicators such as follicle stimulating hormone (FSH) and anti-Mullerian hormone (AMH) each have advantages and disadvantages [9,10]. Therefore, it is of great practical significance to elucidate the physiological and pathological regulatory mechanisms of ovarian aging and explore its potential molecular regulatory targets.

Post-translational modification (PTM) is an important functional regulation pattern in vivo, including acetylation, methylation, ubiquitination, glycosylation, lipidylation and phosphorylation [11]. Currently, several research groups have demonstrated that PTMs also dynamically change during the aging process of different model organisms and cell types, and that changes in specific PTMs can prolong the life span of model organisms [12,13,14]. In addition, a large number of studies have shown that PTMs, vital molecular mechanisms regulating cellular functions, are associated with the occurrence of normal female reproductive events, such as stem cell differentiation, oogenesis, ovulation, hormone secretion, and other biological processes [15,16]. In particular, the histone methylation of H3K4, H3K36, and H3K79 promotes transcription, while methylation of H3K9, H3K27, and H4K20 suppresses transcription [17]. Hong-Thuy Bui et al. [18] found that the ninth lysine of histone H3 is modified by methylation when the follicle develops to the sinusoid stage during the growth stage, and confirmed that the histone methylation that forms during the oocyte growth stage is stable during oocyte maturation and activation. Furthermore, the expression level of H3K4 dimethylation in phase MⅡ oocytes of the vigorous ovarian functioning group was higher than that of the aging group, and the lifespan of worms and *Drosophila* was significantly shortened when the H3K4 demethyase-retinol-binding protein 2 (RBP 2) was deficient [19]. Therefore, PTMs also play an important role in the regulation of ovarian function, but there are few studies on the PTMs of many novel proteins in the female reproductive system. More importantly, whether changes in PTMs are associated with physiological ovarian aging remains unclear.

In this study, the level of histone modifications and the global profiles of PTMs (especially newly identified, metabolism-related lysine modifications) were examined during physiological ovarian aging in mice. Furthermore, we screened for differentially expressed PTM-related regulatory enzymes to evaluate whether gene expression and PTMs are associated with physiological ovarian aging.

## 2. Results

### 2.1. Ovarian Function of 3M, 12M, and 17M Mice Showed Gradual Decline during the Physiological Aging Process

To confirm follicle depletion in the 3M, 12M, and 17M mice, we performed histological analysis of the ovaries. As seen in Figure 1A–D, as the age of the mice increased, the number of primordial and mature follicles was reduced, while the number of atretic follicles increased gradually (*p* < 0.05, *p* < 0.01, and *p* < 0.001). Moreover, the corpus luteum level was highest in 17M mice (*p* < 0.001). The mature follicles showed the most significant difference in 12M and 17M mice (*p* < 0.001). The coefficient of ovary weight decreased between 3M and17M (Figure 1E, *p* < 0.05, *p* < 0.01, and *p* < 0.001).

We used RT-PCR assays to detect AMH and MVH expression in the ovaries of 3M, 12M, and 17M mice (*n* = 10). The results showed that both of the relative mRNA expressions of AMH and MVH decreased significantly as mice aged (Figure 1F,G, *p* < 0.01, *p* < 0.001). On the contrary, the aging marker genes p16 and p53 showed a significant increase among 12M and 17M mice compared with the 3M mice (Figure 1H,I, *p* < 0.01, *p* < 0.001). These data suggest that the ovarian function of 3M, 12M, and 17M mice gradually decreases with age, which may reflect the different functional states of the ovary in the physiological aging process (ovarian function exuberant stage-ovarian function decline stage-ovarian function failure stage).

### 2.2. Ovarian Aging Is Associated with Altered Ovarian Levels of 14 PTMs and Newly Identified Metabolism-Related Lysine Modifications

We used 18 pan-antibodies to test the changes of PTMs in the global protein during the ovarian aging of mice (*n* = 10). The results showed that the Kbhb (lysineβ-hydroxybutyryllysine), Khib (lysine 2-hydroxyisobutyryllysine), Kglu (lysineglutaryllysine), Kmal (lysinemalonyllysine), Ksucc (lysinesuccinyllysine), Kcr (lysinecrotonyllysine), Kbu (lysinebutyryllysine), Kpr (lysinepropionyllysine), SUMO1 (SUMO1 modification), ub (ubiquitination), P-Typ (phosphorylation), and 3-nitro-Tyr (nitro-tyrosine) increased significantly as mice aged, respectively. Meanwhile, the Kme2 (lysinedi-methyllysine) and Kac (lysineacetyllysine) were highest in the 3M mice and lowest in 12M mice. However, there was no effect on the Kme3 (lysinetri--methyllysine), Klac (lysinelactyllysine), Kbz (lysinebenzoyllysine), or O-GlcNA (O-GlcNAcylation) (Figure 2, *p* < 0.05, *p* < 0.01, *p* < 0.001).

### 2.3. Histone Modification Levels of Kme3 and Ubiqutin, Kpr, and O-GlcNA Were Altered during Ovarian Aging

To identify whether PTMs of histones are involved in regulating gene expression in ovarian aging, we extracted the histones from the ovaries of 3M, 12M, and 17M mice (*n* = 10). In this study, the levels of 15 PTMs, including Kme2, Kme3, Kac, Klac, Kbhb, Khib, Kglu, Kmal, Ksucc, Kcr, Kbz, Kbu, Kpr, ubiquitin, and O-GlcNA, were evaluated by western blot using a pan-antibody during ovarian aging (Figure 3 and Figure 4). The screening showed that Kme3 of histone lysine was up-regulated progressively and significantly with increasing age (Figure 3A, *p* < 0.001), H4 ubiquitination was obviously higher in the 12M and 17M mice than 3M (Figure 3D, *p* < 0.001), whereas the Kpr and O-GlcNA level in 17M was significantly decreased compared with the level in 3M mice (Figure 3B,C, Appendix A, *p* < 0.01).

### 2.4. The Expression of Regulatory Enzyme Genes including TIP60, P300, PRDM9, KMT5B, and KMT5C Increases during Ovarian Aging

The main functions of histones and global profiles of PTMs are regulated by serial transferases and demethylases. Given the above results, we next assessed whether this is related to the changes in expression of their regulatory enzymes. As shown in Figure 5, the expression of TIP60, P300 PRDM9, KMT5B, and KMT5C genes was measured by qPCR, and the results showed an increase from 3M to 17M female mice by qPCR (*n* = 10, Appendix A, *p* < 0.05). Furthermore, the results show that the increase in PTM level was positively correlated with the up-regulation of PTMs, regulating enzymes during ovarian aging in mice.

## 3. Discussion

Ovarian aging is mainly affected by complex phased gene expression and transcription factor regulatory networks [1,2,3,4,5,6,7,8,9,10]. Due to high heterogeneity and a complex etiology, the mechanism of ovarian aging is far from clear. Our morphometric analysis of the ovaries showed that mature follicles make up most of the total follicles in 3M mouse ovaries. With further growth and differentiation, primordial follicles are gradually replaced by different stages of follicles. Moreover, qPCR was used to detect the changes in the expression of reproductive function-related genes and senescence marker genes in the ovaries of mice of different ages, which confirmed that 3M, 12M, and 17M mice are able to accurately reflect the different functional states of the ovaries in the physiological aging process.

In the present study, we measured PTMs in the ovaries of 3M, 12M, and 17M mice during the physiological aging process. The results showed that the modification level of PTMs and the expression levels of regulatory enzyme genes showed clear changes in the ovarian hypofunction group (12M) and the ovarian failure group (17M) compared with the adolescent group (3M), suggesting that changes in PTMs are associated with physiological senescence of the ovary. Most importantly, we first used 18 types of PTM pan-antibodies to detect changed PTMs during the global protein level in the physiological ovarian aging process. The results showed that the modification levels of three classical PTMs (Kme2, Kac, and P-Typ), nine novel PTMs (Kbhb, Khib, Kglu, Kmal, Ksucc, Kcr, Kbu, Kpr, and 3-nitro-Tyr), and the expression of regulatory enzymes, including p300 and TIP60, clearly increased. The level of glycolysis decreases in the p300 gene knockout group in HCT116 cells, which reveals that loss of p300 causes a decrease in 2-hydroxyisobutyrylation in glycolytic enzymes, thereby inhibiting glycolysis [20]. However, the results obtained in the present study revealed that 2-hydroxyisobutyrylation and the expression of p300 increase significantly, indicating that increasing p300-dependent 2-hydroxyisobutyrylation can promote glycolysis in order to replenish the consumption of glucose during the aging process. As we known, the concentration of amino acids, including glutamic acid and threonine increased as age increased [21], and our study demonstrated that ubiquitylation and sumolytion increase from 3M to 17M in mice, which suggested that accumulation of amino acids was accompanied by acceleration of protein degradation with increased age in mice. Previous research has shown that the increase of 2-hydroxyisobutyrylation leads to up-regulated expression of related genes for immune regulation, suggesting that 2-hydroxyisobutyrylation may be involved in the metabolic regulation [22]. However, an increase in 2-hydroxyisobutyrylation is also observed in our study, indicating that it may be associated with immune regulation in ovarian senescence. Studies have also been reported that increasing crotonylation may affect cell metabolism, which can promote embryonic stem cells to self-renew in mice [23,24]. In the present study, the level of crotonylation increases during senescence, suggesting that apoptosis of oocytes and granulosa cells may be associated with an increase in crotonylation. Previous research has shown that the mitochondria are rich in abundant proteins, such as malonylation and succinylation, and its dysfunction can trigger oxidative damage and induce ovarian senescence [25], which were consistent with the increase of malonylation and succinylation in ovarian senescence in our research. Currently, we predicted that succinylation may regulate ovarian function by altering the function of mitochondria during ovarian senescence. Moreover, it has been reported that deglutarylation has an anti-oxidative effect on maintain intracellular redox homeostasis [26]. Our present study observed that an increase of glutarylation may mediate the reduction of antioxidative mechanisms by trigger the oxidative stress response to regulate ovarian senescence. To date, the physiological roles of butyrylation and benzoylation in ovarian senescence still remain unclear; therefore, our study provides a new approach for the exploration of novel PTMs of ovarian physiological functional proteins.

Histones are the protein components of the nucleosome, which form the basic architecture of eukaryotic chromatin. The N terminus of histones H2A, H2B, H3, and H4 can undergo many types of post-translational modification, including methylation, phosphorylation, acetylation, and ubiquitination [27]. Histones PTMs have become a research hotspot, because they not only participate in the key molecular network of ovarian physiological function regulation, but also may be an important pathological mechanism of common ovarian diseases. Many studies have indicated that histone modification is an effective indicator of production stability and follicular development [28,29]. However, the histones PTMs in a physiological aging population were analyzed using single-cell chromosome sequencing, and the results showed that epigenetic changes of histones play a significant role in the genetic aging of immune cells [30]. Importantly, the role of histone PTMs during reproductive aging in female mammals has not been reported. As shown in Figure 3, the modification level of Kpr and O-GlcNA were obviously higher in the 17M mice than 3M, which indicated that metabolic energy, cell-mediated immunity may regulate ovarian functions by specific enzymes. The novel study revealed that the ubiquitination of histones is involved in the spermiogenesis for male reproduction [31], and H4 ubiquitination was obviously higher in the 12M and 17M mice than 3M during our research, which was relevant to cell apoptosis and aging in female reproduction. Importantly, here we report the discovery of one novel, in vivo lysine modification in histone3, lysinetri-methyllysine. Our results showed that the level of H3 methylation is up-regulated significantly with the increase of age compared to the 3M group, indicating that H3 methylation may play an important role in regulating physiological function during ovarian senescence (Figure 3). The main functions of H3 methylation involved the maintenance of the chromatin structure surrounding the centromere and protection of genome stability through transposon silencing, which was regulated by methyl-transferase and demethylase [32]. Therefore, we selected three methyltransferases, PRMD9, KMT5B, and KMT5C, which showed a progressive increase in protein abundance with increasing age. Additionally, other transferases and demethylases showed no change during physiological ovarian aging in mice (Appendix A). Previous research has shown that KMT5C promotes senescence via miR-29-induced loss of H4K20me3 [33], and KMT5C become a potential biomarker and serves as an early identification of the risk of a variety of malignant diseases and prognosis assessment, including ovarian cancer and breast cancer [34,35,36]. However, it is not clear how important KMT5C is in the ovarian aging process, and the specific function network of KMT5C is worthy of further exploration.

Meanwhile, our sequencing analysis showed that the level of histone glycosylation and propionylation only decreased significantly in the 17M group, and no obvious change was observed in the 12M group. A previous study demonstrated that histone glycosylation is strongly associated with the metabolism of amino acids, fatty acids, and energy [37], while lysine propionylation of histones as a novel PTM, is mainly related to the global regulatory network and cellular stress response. Taken together, these results suggested that metabolic dysregulation may appear at the ovarian failure stage, including premature ovarian failure (POF) and polycystic ovarian syndrome (PCOS).

## 4. Conclusions

In conclusion, this study demonstrates that female reproductive function-related genes and the modification of Kme3, ubiqutin, Kpr, and O-GlcNA, as well as global levels of PTMs, including metabolism-related modification, phosphorylation, and nitro-tyrosine, are altered during physiological aging in the mouse ovaries, indicating the diverse and complicated regulation of gene expression and PTMs in age-induced female reproductive dysfunction. In addition, we found that KMT5C also shows a significant increase in the ovaries of mice treated with cyclophosphamide (120 mg/kg) and busulfan (12 mg/kg) at 2M, which were selected as pathological ovarian aging models. Together, these results suggested that KMT5C and Kme3 as-mediated epigenetic pathways may play a key role in the ovarian aging process, and their specific role is worthy of further study and exploration.

## 5. Materials and Methods

### 5.1. Animals

Three-month-old, twelve-month-old, and seventeen-month-old KM female mice were housed at the Center of Experimental Animals, Nanchang University. The weights of the mice and ovaries were measured and recorded on the last day of treatment. All mice were housed in 12-h dark and 12-h light environments with a suitable temperature and an adequate supply of food and water. All of the mice were purchased from the Experimental Animal Science and Technology Center of Jiangxi University of Traditional Chinese Medicine (Permit Number: SCXK2018-0003, Appendix A) and treated humanely and with regard for the alleviation of suffering.

### 5.2. Histological Analysis of Ovarian Tissue and Ovarian Follicle Count

After the mice were killed, both ovaries were quickly removed to ensure the freshness of the tissues. Attention was paid to avoid squeezing the ovaries during clamping, and the integrity of the ovaries was ensured. After removal, the ovaries were placed in sterilized physiological saline (Appendix A) solution to remove the surface blood, and the surface water of the ovaries was dried with absorbent paper. For the evaluation of follicles, ovaries were fixed in 4% paraformaldehyde and prepared as paraffin blocks; 5-mm sections were obtained serially and stained with hematoxylin-eosin (HE). The ovarian follicles were counted as described previously [38,39].

### 5.3. Quantitative Real-Time PCR

The total RNA from the different stages of female mice and treated ovarian surface epithelium was extracted using TRIzol reagent (Invitrogen, Waltham, MA, USA) and cDNA was synthesized using a PrimeScript RT reagent Kit with gDNA Eraser (TaKaRa, Kusatsu, Japan), according to the manufacturer’s instructions. The mean values were used for the determination of the mRNA expression levels using the comparative ΔCq (ΔCt) method, according to the equation 2^−ΔΔCq^ (2^−ΔΔCt^); GAPDH was used as the reference gene (Appendix A). All primers were obtained from GENEWIZ, Inc. (Suzhou, China). The RT-qPCR samples were assayed in triplicate to achieve good reproducibility.

### 5.4. Western Blot Analysis

Total protein was extracted from different ovarian tissues with RIPA lysis solution (Beyotime, P0013C). Histones were extracted from the different stages of mice using NETN Lysis Buffer (100 mM NaCl, 20 mM tris-Cl (pH 8.0), 0.5 mM EDTA, and 0.5% (*v*/*v*) NP-40). In addition, PR-619 (50 μM) was used during the extraction of ubiquitin. Equal amounts (30 μg) of protein were separated by 12% sodium dodecyl sulfate polyacrylamide gel electrophoresis (SDS-PAGE) and then electrophoretically transferred onto the PVDF membranes (Millipore Corp., Bedford, MA, USA) and processed per the antibody manufacturer’s instructions. After incubating with the primary antibodies against PTMs, the filter was visualized using anECL detection kit (Pierce; Thermo Fisher Scientific, Waltham, MA, USA). Analysis of PTMs in the different stages of female mice was conducted by quantifying the grey value of target bands detected by the corresponding antibodies normalized to those detected by GAPDH for anti-pan-PTM (Appendix A) antibodies and by histones (histone 3 and 4) for anti-histone modification antibodies using Image J software (version 1.44, National Institutes of Health, Bethesda, MD, USA). Each experiment was repeated at least three times.

### 5.5. Immunoprecipitation

After cells were lysed with RIPA buffer (Beyotime, P0013C) containing PMSF (DINGGUO, WB0181) and cocktail, the cell lysates were precleaned with protein G agarose (Roche, 11, 243, 233, 001) at 4 °C for 1 h, after which the supernatant was incubated with the indicated antibodies and protein G agarose beads at 4 °C overnight. On the second day, immunocomplexes combined with beads were washed with lysis buffer, followed by western blot.

### 5.6. Statistical Methods

All analyses were performed using GraphPad Prism 5.01 (GraphPad Software, Inc., San Diego, CA, USA). The statistical comparisons among different groups were expressed as the mean ± standard error (SEM) and the data from the experiments were analyzed by ANOVA. The threshold of *p* < 0.05 was considered significant; *p* < 0.01 and *p* < 0.001 were considered extremely significant.

## Figures and Tables

**Figure 1 ijms-23-00002-f001:**
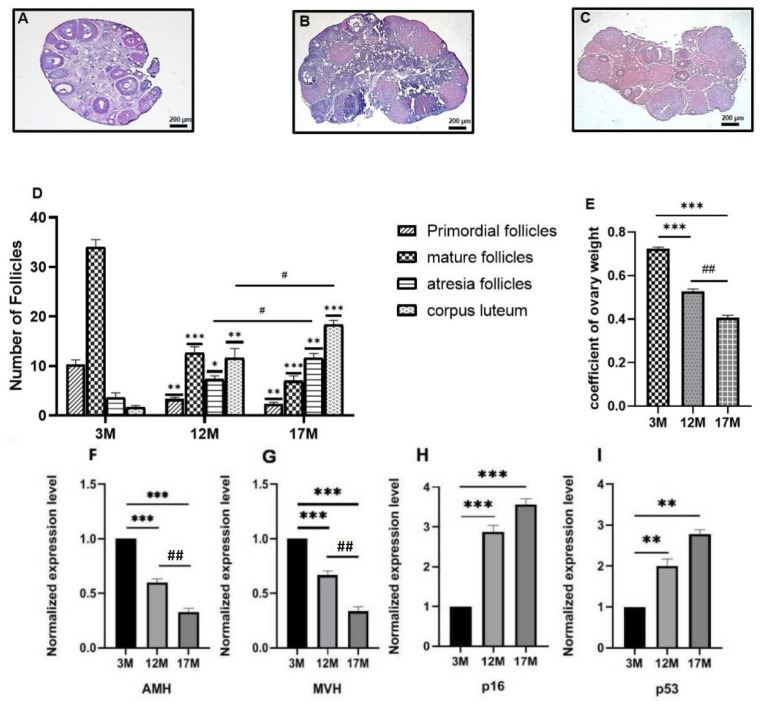
Changes of ovarian morphology and gene expression related to follicular development in mice of different ages. The dynamic changes of follicles during physiological ovarian aging in the mouse ovaries. (**A**–**C**) The detection of ovarian follicle depletion by HE-staining. (**D**) The number of primordial and mature follicles is reduced with increased age, but atretic follicles increased gradually. (**E**) The coefficient of ovary weight showed decreased from 3M to 17M. (**F**–**I**) Total RNAs from the ovaries of different age in mice were isolated and the expression of genes encoding reproductive function-related proteins was examined by quantitative real-time PCR using the 2^−ΔΔCt^ method. the relative mRNA expression of AMH and MVH were decreased significantly as mice aged; the aging marker genes p16 and p53 showed a significant increased among 12M and 17M mice compared with the 3M mice. The results are presented as the mean ± SD. * *p* < 0.05, ** *p* < 0.01, *** *p* < 0.001, compared with the 3M group. # *p* < 0.05, ## *p* < 0.01, compared with the 12M group. Scale bars represent 200 μm.

**Figure 2 ijms-23-00002-f002:**
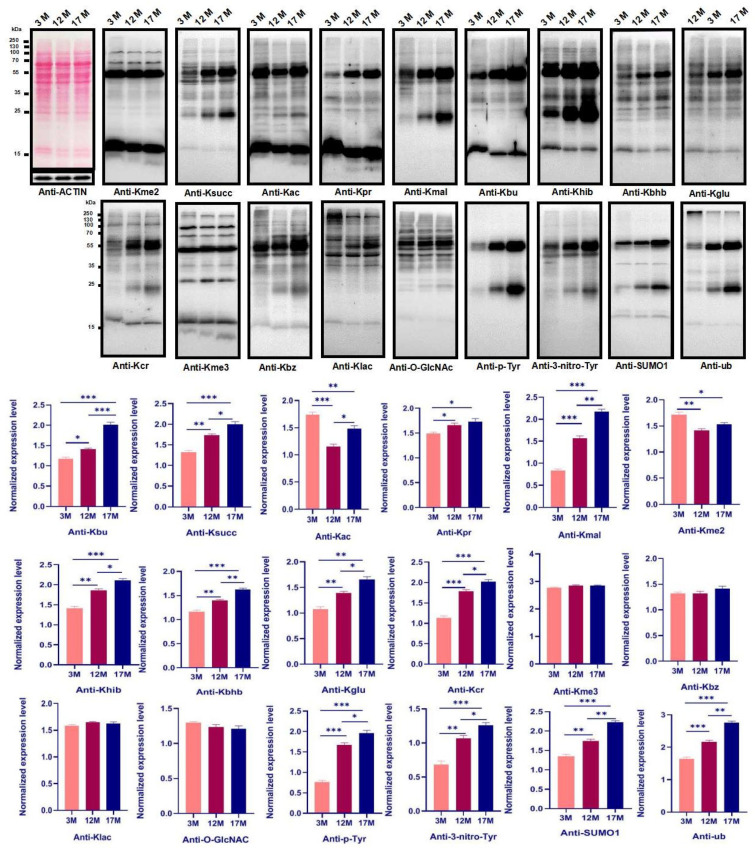
The global profiles of post-translational modifications (PTMs) during physiological ovarian aging in the mouse ovaries. Ovaries proteins in different age of mice were isolated and the global level of PTMs was examined by western blot and analyzed by quantifying the sum of the gray values of detected bands normalized to those of b-ACTIN using Image J software (version 1.44, National Institutes of Health, Bethesda, MD, USA). Kme2, lysinedi-methyllysine; Kme3, lysinetri--methyllysine; Kac, lysineacetyllysine; Kpr, lysinepropionylation; Kmal, lysinemalonylation; Kbu, lysinebutyryllysine; Khib, lysine2-hydroxyisobutyrylation; Kbhb, lysineβ-hydroxybutyryllysine; Kcr, lysinecrotonylation; Ksucc, lysinesuccinylation; Kglu, lysineglutarylation; Kbz, lysinebenzoylation; Klac, lactyllysine; P-Typ, phosphotyrosine; 3-nitro-Tyr, nitro-tyrosine; SUMO1, SUMO1 modification; ub, ubiqutin; O-GlcNAc, O-GlcNAcylation. Bar: mean ± SEM, * *p* < 0.05, ** *p* < 0.01, *** *p* < 0.001.

**Figure 3 ijms-23-00002-f003:**
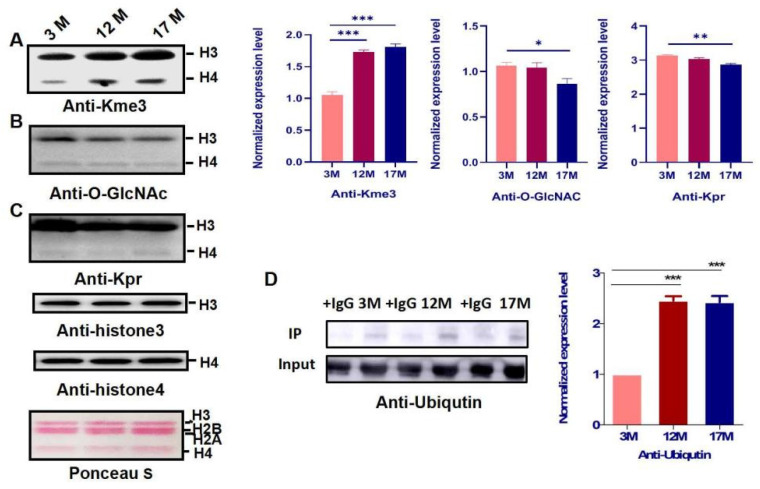
The histone modification level of Kme3 and ubiqutin, Kpr, and O-GlcNA of histone were altered during ovarian aging. (**A**–**C**) Ovaries histones in different age of mice were isolated and the level of Kme3, Kpr, and O-GlcNA modifications were examined by western blot and analyzed by quantifying the gray value of target bands normalized to those detected by Kme3, Kpr, and O-GlcNA using Image J software. (**D**) ubiqutin was examined by IP methord and and analyzed by quantifying the gray value of target bands normalized to those detected by ubiqutin using Image J software. Bar: mean ± SEM, * *p* < 0.05, ** *p* < 0.01, *** *p* < 0.001.

**Figure 4 ijms-23-00002-f004:**
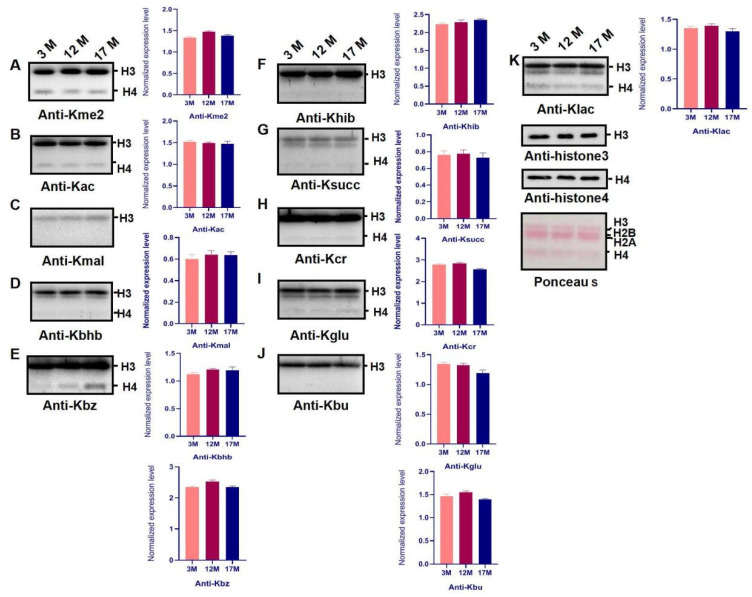
The modification changes of PTMs in histone during ovarian aging of mice. (**A**–**K**) Ovaries histones in different age of mice were isolated and the level of lysinedi-methyllysine (Kme2), lysineβ-hydroxybutyryllysine (Kbhb), lysine2-hydroxyisobutyryllysine (Khib), lysineglutaryllysine (Kglu), lysinemalonyllysine (Kmal), lysinesuccinyllysine (Ksucc), lysinecrotonyllysine (Kcr), lysinebenzoyllysine (Kbz), lysinebutyryllysine (Kbu), lysineacetyllysine (Kac), and lysinelactyllysine (Klac). Modifications were examined by western blot and analyzed by quantifying the gray value of target bands normalized to those detected by PTMs using Image J software (version1.44, National Institutes of Health, MD, USA). Bar: mean ± SEM.

**Figure 5 ijms-23-00002-f005:**
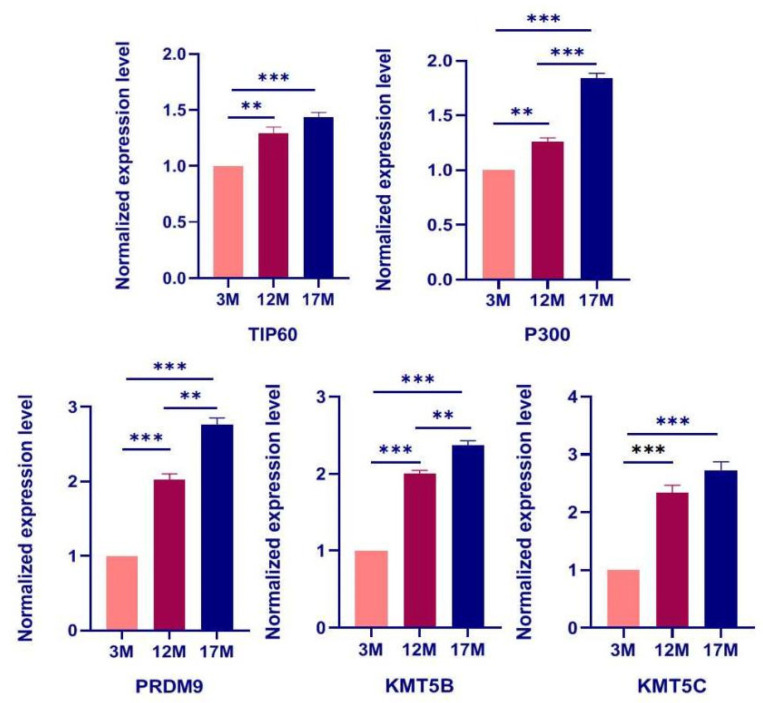
The expression of related regulatory enzymes was examined by quantitative real-time PCR using the 2^−ΔΔCt^ method. TIP60, Tat-interaction protein; P300, E1A binding protein p300; PRDM9, PR domain zinc finger protein 9; KMT5B, Lysine methyltransferase 5B; KMT5C, Lysine methyltransferase 5C. Bar: mean ± SEM, ** *p* < 0.01, *** *p* < 0.001. Student *t*-test.

## Data Availability

The data presented in this study are available on request from the corresponding author.

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
