# Peer review of "Physiological Ovarian Aging Is Associated with Altered Expression of Post-Translational Modifications in Mice"

_ijms, 2021, doi:10.3390/ijms23010002_

Round 1

Reviewer 1 Report

Author have addressed by reviewers questions. I dont have any further questions ask except minor update on section 2.2   PTM- 18 antibody western blot were presented with out pointing the specific band on the blot. Each blot has multiple nonspecific bands. 

Author Response

Comments and Suggestions for Authors
1

I recommend the acceptance of the revised manuscript in its present form.

Authors replied: Thanks for your comment.

Reviewer 2 Report

I recommend the acceptance of the revised manuscript in its present form.

Author Response

Comments and Suggestions for Authors
2

Author have addressed by reviewers questions. I dont have any further questions ask except minor update on section 2.2   PTM-18 antibody western blot were presented with out pointing the specific band on the blot. Each blot has multiple nonspecific bands. 

Authors replied: Thanks for your comment. The 18 pan-antibodies were used to test the changes of PTMs in the global protein, which can identify all modified protein, not referring to a protein, all stripes in the figure2 occurred modification.

This manuscript is a resubmission of an earlier submission. The following is a list of the peer review reports and author responses from that submission.

Round 1

Reviewer 1 Report

The authors  used the mouse model to study the global profiles of post-translational modifications, as well as the level of histone modifications and related regulatory enzymes during ovarian aging by quantitative real-time PCR and western blot, respectively. They found altered regulated enzymes and post-translational modifications to be altered during physiological ovarian aging. These findings open new ways to study events upstream of hormonal changes occurring in human ovaries when the aging process has already become irreversible.

The study is wel-designed, the methods are adequate, and the interpretations reflect the findings. The manuscript is well-written and clear.

I recommend acceptance of the paper in its present form.

Minor points

  1. Abstract, line 11: Please remove the word “displayed”.
  2. The authors may wish to contrast their findings with those concerning the expression of major factors involved in human ovarian aging (reviewed in Tesarik J, et al. Ovarian Aging: Molecular Mechanisms and Medical Management. Int J Mol Sci. 2021 Jan 29;22(3):1371. doi: 10.3390/ijms22031371).

Author Response

reviewer1:

Minor points

  1. Abstract, line 11: Please remove the word “displayed”.

Reply: I have removed the word "displayed" in abstract.

  1. The authors may wish to contrast their findings with those concerning the expression of major factors involved in human ovarian aging (reviewed in Tesarik J, et al. Ovarian Aging: Molecular Mechanisms and Medical Management. Int J Mol Sci. 2021 Jan 29;22(3):1371. doi: 10.3390/ijms22031371).

Reply: Thank you for your suggesting. I have modified the reference 9th for Tesarik, J.; Galán-Lázaro, M.; Mendoza-Tesarik, R. Ovarian Aging: Molecular Mechanisms and Medical Management. Int. J. Mol. Sci. 2021, 22,1371.

Reviewer 2 Report

Physiological ovarian aging is associated with altered expression of post-translational modifications in mice.

Mingli Wei`s manuscript described the modulation of post-translational modification in ovarian aging mice.

Beside the interesting approach, the manuscript has a deep lack of knowledge about how to detect and analyzed post-translational modifications to proteins.

Major concerns:

The Fig2 illustrates the global profiles of post-translational modifications in the analyzed samples. No representative western blots are showed to support the Fig 2 data.

It is confusing and the reader cannot understand the logic of why the main message of PTM in the histones were put in the Supplementary information. What is the correlation between variation in PTM observe at the level of total protein and in the histone? A full panel of uncropped blots with the indicated molecular weights are required.

No experiment of Histone purification by immunoprecipitation and probed with specific PTM antibodies have been done. Because of Ub and SUMO have almost the same molecular weight of the histones and they are also highly detected in the nuclei, how the Authors can avoid a simple contamination problem? No molecular weights are indicated in the blots.

Specific inhibitors of SUMO and Ub deconjugases were not used in the lysis buffer as described in the Materials and Methods chapter; the commercial protease inhibitor cocktails or tablets are not sufficient to block the mentioned deconjugases, suggesting that the SUMO and Ub PTMs maybe were not preserved during sample manipulation.

There is no indication about how many mice were used in the experiments per group, how the statistics analysis has been done?

A massive revision of typos mistakes and English proofreading must be considered.

Author Response

reviewer2:

Major concerns:

  1. The Fig2 illustrates the global profiles of post-translational modifications in the analyzed samples. No representative western blots are showed to support the Fig 2 data.

Replay:  I am very sorry for this point. I have add the figure about western blots in the fig 2 data.

  1. It is confusing and the reader cannot understand the logic of why the main message of PTM in the histones were put in the Supplementary information. What is the correlation between variation in PTM observe at the level of total protein and in the histone? A full panel of uncropped blots with the indicated molecular weights are required.

Replay: I am very sorry for this point. The main message of PTM in the histones were put in the figure 4. Histones are the protein components of the nucleosome, which forms the basic architecture of eukaryotic chromatin. The N terminus of histones H2A, H2B, H3, and H4 can undergo many types of post-translational modification, including methylation, phosphorylation, acetylation, sumoylation and ubiquitination. We found that PTM observe at the level of total protein and in the histone have the general similarities, but also because of its uniqueness, but also have different additional features. Because data is very informative, we are doing these study, which will be put in other paper.

  1. No experiment of Histone purification by immunoprecipitation and probed with specific PTM antibodies have been done. Because of Ub and SUMO have almost the same molecular weight of the histones and they are also highly detected in the nuclei, how the Authors can avoid a simple contamination problem? No molecular weights are indicated in the blots.

Replay: Thanks for your comment. The histone purification were used by acid extraction. The company reply the SUMO and Ub antibody are different,

  Anti-SUMO1 Mouse mAb

货号:PTM-1110

                           Anti-Ubiquitin Mouse mAb (NT)

                              货号:PTM-1107

  1. Specific inhibitors of SUMO and Ub deconjugases were not used in the lysis buffer as described in the Materials and Methods chapter; the commercial protease inhibitor cocktails or tablets are not sufficient to block the mentioned deconjugases, suggesting that the SUMO and Ub PTMs maybe were not preserved during sample manipulation.

Replay: Thanks for your comment.  The company reply us to specific inhibitors of PR-619(50 uM) during the the extract of Ub, N-ethylmaleimide(10 mM) during the extract of sumo.

  1. There is no indication about how many mice were used in the experiments per group, how the statistics analysis has been done?

Replay:  I am very sorry for this point. In our experiments, we used 10 mice in each experiment per group.

  1. A massive revision of typos mistakes and English proofreading must be considered.

Replay: Thanks for your comment. This revision has been revised at https://www.proof-reading-service.com/en/received/ and checked by our a native English-speaking colleague.

Reviewer 3 Report

In this manuscript, author studied physiological ovarian aging associated with altered expression of PTM in ovarian mouse model by qpcr and western analysis. PTM in aging in general, protein PTM  takes place as a result of either modifying enzymes related to posttranslational processing (eg. glycosylation) or signaling pathway activation, such as phosphorylation. PTM patterns are known to be affected by disease conditions similar to the dysregulation of PTM is associated with the aging process.in this condition, both enzymatic and nonenzymatic PTMs can undergo age-related alterations.  Manuscript is very primitive stage with limited methodology screening analysis. Overall, Well-written manuscript, Despite some weaknesses by omission, the core of the work here is a strong foundation and is a relatively complete, if more modest, story.

some of the minor points to be noted

a. need to update IACUC study protocol , number of animals per group

b.  observed follicle count variation based on aging, did you addressed PTM markers on  sub cellular localization changes during aging by IHC-IF

c. HAT and HDAC are associated to  protein acetylation and histone modification and also part of PTM, do you have any data to support this because of  mentioning broad title PTM

d. specify vendor catalog # details of all the antibody used in this study

Author Response

reviewer3:

some of the minor points to be noted

  1. need to update IACUC study protocol , number of animals per group

Reply: Thanks for your comment. I have updated the IACUC study protocol in supplementary files. Furthermore, in our experiments, we used 10 mice in each experiment per group.

  1. observed follicle count variation based on aging, did you addressed PTM markers on  sub cellular localization changes during aging by IHC-IF

Reply: I am very sorry for this point. These results of IHC-IF during some PTM markers such as Kme3, Kpr, and O-GlcNA, which are the experiments we're doing right now ,and will be published as a follow-up article.

for example:

  1. HAT and HDAC are associated to  protein acetylation and histone modification and also part of PTM, do you have any data to support this because of  mentioning broad title PTM

Reply:  The histone of acetylation has been modified during ovarian aging. Currently, we has some datas to support the correlation during HAT , HDAC and acetylation. Thank you for your propose, We'll think about it in next researches.

  1. specify vendor catalog # details of all the antibody used in this study

Reply: I have add the specify vendor catalog # details of all the antibody in supplementary files.

Round 2

Reviewer 2 Report

I appreciate the effort in the reviewing of the manuscript, but I still have major concerns about how the experiments have been performed.

  1. Regarding my previous point 3: 

3.No experiment of Histone purification by immunoprecipitation and probed with specific PTM antibodies have been done. Because of Ub and SUMO have almost the same molecular weight of the histones and they are also highly detected in the nuclei, how the Authors can avoid a simple contamination problem? No molecular weights are indicated in the blots.

Authors replied: Thanks for your comment. The histone purification were used by acid extraction. The company reply the SUMO and Ub antibody are different,

Authors pasted the data sheet information regarding the antibodies. The reviewer is perfectly conscious about the different specificity of the described antibodies, but I wanted to see immunoprecipitation experiments where Histones were pulled down with the specific antibodies and then probed with anti-SUMO and anti-Ubiquitin antibodies. The reason is that, in the histone lysate there is the possiblity to have contamination of free SUMO and free Ubiquitin since both moieties are present in the nuclei. The ONLY way to determine the histone PTM mediated by SUMO or Ub is with immuniprecipitation.

  1. 2. Regarding my previous point 4: 
  2. Specific inhibitors of SUMO and Ub deconjugases were not used in the lysis buffer as described in the Materials and Methods chapter; the commercial protease inhibitor cocktails or tablets are not sufficient to block the mentioned deconjugases, suggesting that the SUMO and Ub PTMs maybe were not preserved during sample manipulation.

Authors replied: Thanks for your comment. The company reply us to specific inhibitors of PR-619(50 uM) during the the extract of Ub, N-ethylmaleimide(10 mM) during the extract of sumo.

Beside this clarification, in the revised version of the manuscript there is no indication that these inhibitors were added to the lysis buffer. Were they present or not? The reviewer doesn’t understand the role of the Company in this concept.

  1. 3. Regarding my previous point 5: 
  2. There is no indication about how many mice were used in the experiments per group, how the statistics analysis has been done?

Authors replied: I am very sorry for this point. In our experiments, we used 10 mice in each experiment per group.

The number was not written in the new version of the manuscript.

  1. 4. Regarding my previous point 6: 
  2. A massive revision of typos mistakes and English proofreading must be considered.

Authors replied: Thanks for your comment. This revision has been revised at https://www.proof-reading-service.com/en/received/ and checked by our a native English-speaking colleague.

I understand the English proof-reading corrections, unlikely typing mistakes are still detected along the manuscript, for example.

  • In all graphs the Y axes has “Noralized expression level
  • Fig legend 2: “Ovaries proteins in different age of mice were isolated and the global level of PTMs was examined by western blot and analyzed by quantifying the sum of the gray values of detected bands (black arrows)” . Where are the black arrows?
  • Page 5 last line: (Figure. 3, Figure. S14 p< 0.05). What is “Figure. S14” referred to?
  • Page 8. Any figure 45?

This suggests a poor revision in all the parts of the second manuscript version.

Minor comment:

In the Supplementary files, Figure S2, Authors copied a photo of IACUC study protocol. Unlikely the pic is in Chinese, and it is not readable from the no Chinese scientific community and no suitable for an international science journal where the official language is English. Authors should remove or provide a translation.

Author Response

Comments and Suggestions for Authors

I appreciate the effort in the reviewing of the manuscript, but I still have major concerns about how the experiments have been performed.

  1. Regarding my previous point 3: 

3.No experiment of Histone purification by immunoprecipitation and probed with specific PTM antibodies have been done. Because of Ub and SUMO have almost the same molecular weight of the histones and they are also highly detected in the nuclei, how the Authors can avoid a simple contamination problem? No molecular weights are indicated in the blots.

Authors replied: Thanks for your comment. The histone purification were used by acid extraction. The company reply the SUMO and Ub antibody are different,

Authors pasted the data sheet information regarding the antibodies. The reviewer is perfectly conscious about the different specificity of the described antibodies, but I wanted to see immunoprecipitation experiments where Histones were pulled down with the specific antibodies and then probed with anti-SUMO and anti-Ubiquitin antibodies. The reason is that, in the histone lysate there is the possiblity to have contamination of free SUMO and free Ubiquitin since both moieties are present in the nuclei. The ONLY way to determine the histone PTM mediated by SUMO or Ub is with immuniprecipitation.

Reply: Thank you for your suggestion. This was an oversight in our experiment, and we really should have allowed for this to happen. But since the editor only gave me two days to revise this manuscript, I didn't have time to catch up on the immunoprecipitation experiments. The results of SUMO and Ub have been deleted in this article so as not to mislead the reader. Would you please understanding!

  1. 2. Regarding my previous point 4: 
  2. Specific inhibitors of SUMO and Ub deconjugases were not used in the lysis buffer as described in the Materials and Methods chapter; the commercial protease inhibitor cocktails or tablets are not sufficient to block the mentioned deconjugases, suggesting that the SUMO and Ub PTMs maybe were not preserved during sample manipulation.

Authors replied: Thanks for your comment. The company reply us to specific inhibitors of PR-619(50 uM) during the the extract of Ub,N-ethylmaleimide(10 mM) during the extract of sumo.

Beside this clarification, in the revised version of the manuscript there is no indication that these inhibitors were added to the lysis buffer. Were they present or not? The reviewer doesn’t understand the role of the Company in this concept.

Reply: I am so sorry to this point. This company refers to the company from which we buy reagents. N-ethylmaleimide was bought from Med Chem Express(MCE), and PR-619 was bought from Glpbio. In addition, these inhibitors were added to the lysis buffer in the revised version of the manuscript.

  1. 3. Regarding my previous point 5: 
  2. There is no indication about how many mice were used in the experiments per group, how the statistics analysis has been done?

Authors replied: I am very sorry for this point. In our experiments, we used 10 mice in each experiment per group.

The number was not written in the new version of the manuscript.

Reply: I am so sorry to this point. I have been modified this point in the manuscript.

  1. 4. Regarding my previous point 6: 
  2. A massive revision of typos mistakes and English proofreading must be considered.

Authors replied: Thanks for your comment. This revision has been revised at https://www.proof-reading-service.com/en/received/ and checked by our a native English-speaking colleague.

I understand the English proof-reading corrections, unlikely typing mistakes are still detected along the manuscript, for example.

  • In all graphs the Y axes has “Normalized expression level
  • Reply: In all graphs the Y axes have been modified for "Normalized expression level"
  •  
  • Fig legend 2: “Ovaries proteins in different age of mice were isolated and the global level of PTMs was examined by western blot and analyzed by quantifying the sum of the gray values of detected bands (black arrows)” . Where are the black arrows?
  • Reply: I am sorry that this is a clerical error of the author and has been deleted in the article. Please look at it.
  •  
  • Page 5 last line: (Figure. 3, Figure. S14p< 0.05). What is “Figure. S14” referred to?

Reply: Thank you for your suggestion. This should be a display error, It should be "Figure 4".

  • Page 8. Any figure 45?
  • Reply: Thank you for your suggestion. This should be a display error, the figure 4 was modified "figure5".

This suggests a poor revision in all the parts of the second manuscript version.

 Minor comment:

In the Supplementary files, Figure S2, Authors copied a photo of IACUC study protocol. Unlikely the pic is in Chinese, and it is not readable from the no Chinese scientific community and no suitable for an international science journal where the official language is English. Authors should remove or provide a translation.

Reply: I have translated the photo as follows:

Animal Use Permit

License No. :SYXK(gan)-2015-0001

Name of the entity: Nanchang University

Legal representative: Changbing Zhou

Facility Address: Medical Laboratory Animal Center, Nanchang University, No. 71 East Yangming Road, Nanchang city

Application: barrier environment

Round 3

Reviewer 2 Report

Comments and Suggestions for Authors

With my regret, also the last manuscript revision (ijms-1370982-peer-review-v3-2(1)) is not adequate for the final acceptance.

The reviewer asked Authors to perform crucial experiments of immunoprecipitations (see comment point 1, revision2). Authors replied that Editor provided only two days for the revision. For this purpose, Authors should have asked a legitimate extension of time to the Editor for the revision, to perform the appropriate IP experiments.

However to avoid this procedure, Authors replied that they have decided to eliminate the SUMO and Ub topics in the article. Unfortunately,

  • In the Abstract SUMO and Ub are still mentioned.
  • In Figure 2 there are blots and analysis about SUMO and Ub (page 4 and 5). More, there are two panels of the same quantification, one with the Y axes “Noralization” and the second with the corrected Y axes “Normalization”. Why?
  • Chapter 2.2 and Figure legend 2, there are SUMO1 and Ub modification discussions.
  • Figure legend 4: SAE1 is a component of SUMO cycle. Since Authors have decided to delete the SUMO and Ub results, what is the purpose to present the variation of the E1 SUMO enzyme (SAE1)? The reviewer/reader is worried about the approach and the scientific message of the revised manuscript version.
  • In the discussion chapter, pages 10-11, SUMO and Ub are mentioned “and the fact that our study shows that ubiquitylation and sumolytion increase from 3M to 17M in mice…”, “The N terminus of histones H2A, H2B, H3, and H4 can undergo many types of post-translational modification, including methylation, phosphorylation, acetylation, sumoylation and ubiquitination.”, as well as in the conclusion Chapter, page 12, “global levels of PTMs, including metabolism-related modification, SUMO1 modification, ubiquitination, phosphorylation, and nitro-tyrosine, are altered during physiological aging in the mouse ovaries” Again, Authors decided to remove the SUMO and Ub concepts, however the reviewer denoted a not accurate revision of the last (ijms-1370982-peer-review-v3-2(1) manuscript version.
  • In Material and Methods, Authors kept “In addition, PR-619(50 uM) during the the extract of Ub, N-ethylmaleimide(10 mM) during the extract of sumo. Is there now a reason to maintain this point when Authors decided to remove SUMO and Ub concepts?
